# Screening and treatment to reduce severe hyperbilirubinaemia in infants in primary care (STARSHIP): a factorial stepped-wedge cluster randomised controlled trial protocol

Berthe A M van der Geest,[1,2] Johanna P de Graaf,[2] Loes C M Bertens,[2] Marten J Poley,[3,4,5] Erwin Ista,[4,6] René F Kornelisse,[1] Irwin K M Reiss,[1] Eric A P Steegers,[2] Jasper V Been,[1,2,7] STARSHIP study group

**To cite:** van der Geest BAM, de Graaf JP, Bertens LCM, *et al*. Screening and treatment to reduce severe hyperbilirubinaemia in infants in primary care (STARSHIP): a factorial stepped-wedge cluster randomised controlled trial protocol. *BMJ Open* 2019;**9**:e028270. doi:10.1136/bmjopen-2018-028270

For numbered affiliations see end of article.

**Correspondence to**
Dr Jasper V Been;
j.been@erasmusmc.nl

## ABSTRACT

**Introduction** Jaundice caused by hyperbilirubinaemia is a physiological phenomenon in the neonatal period. However, severe hyperbilirubinaemia, when left untreated, may cause kernicterus, a severe condition resulting in lifelong neurological disabilities. Although commonly applied, visual inspection is ineffective in identifying severe hyperbilirubinaemia. We aim to investigate whether among babies cared for in primary care: (1) transcutaneous bilirubin (TcB) screening can help reduce severe hyperbilirubinaemia and (2) primary care-based (versus hospital-based) phototherapy can help reduce hospital admissions.

**Methods and analysis** A factorial stepped-wedge cluster randomised controlled trial will be conducted in seven Dutch primary care birth centres (PCBC). Neonates born after 35 weeks of gestation and cared for at a participating PCBC for at least 2 days within the first week of life are eligible, provided they have not received phototherapy before. According to the stepped-wedge design, following a phase of 'usual care' (visual assessment and selective total serum bilirubin (TSB) quantification), either daily TcB measurement or, if indicated, phototherapy in the PCBC will be implemented (phase II). In phase III, both interventions will be evaluated in each PCBC. We aim to include 5500 neonates over 3 years. Primary outcomes are assessed at 14 days of life: (1) the proportion of neonates having experienced severe hyperbilirubinaemia (for the TcB screening intervention), defined as a TSB above the mean of the phototherapy and the exchange transfusion threshold and (2) the proportion of neonates having required hospital admission for hyperbilirubinaemia treatment (for the phototherapy intervention in primary care).

**Ethics and dissemination** This study has been approved by the Medical Research Ethics Committee of the Erasmus MC Rotterdam, the Netherlands (MEC-2017-473). Written parental informed consent will be obtained. Results from this study will be published in peer-reviewed journals and presented at (inter)national meetings.

**Trial registration number** NTR7187.

### Strenghts and limitations of this study

► This study is the first randomised controlled trial (RCT) to assess the (cost) effectiveness of transcutaneous bilirubin measurement as a universal screening tool for potentially severe jaundice, addressing a knowledge gap highlighted in several national guidelines.
► This study is also the first RCT to assess the (cost) effectiveness of phototherapy outside the hospital setting.
► The stepped-wedge design of this study allows all participating primary care birth centres to have introduced both interventions as standard care towards the end of the study, which will aid implementation if (cost) effectiveness of the interventions is demonstrated.
► A limitation is that parents and care providers cannot be blinded to allocation of the interventions.

## INTRODUCTION

In newborns, severe hyperbilirubinaemia may cause brain damage when it is not recognised or is left untreated.[1 2] This spectrum of neurological sequelae is called kernicterus spectrum disorder (KSD), with acute bilirubin encephalopathy and its consequences at the extreme end, and more subtle disorders at the other.[1 3] KSD also encompasses 'classical kernicterus', a chronic, irreversible, neuropathological disorder involving neuromotor dysfunction, auditory disorders and oculomotor impairments.[1–4]

In high income countries, an estimated 1 in 67 000 neonates develop classical kernicterus.[5 6]

Timely recognition of potentially severe jaundice, caused by hyperbilirubinaemia, is essential to prevent KSD.[7 8]

Traditionally, jaundice is identified via visual inspection by maternity care professionals, midwives or parents. If potentially severe jaundice is suspected, total serum bilirubin (TSB) is quantified in blood taken via a skin prick to assess the need for treatment.[9] However, visual inspection of neonatal jaundice is proven to be inaccurate in detecting hyperbilirubinaemia[10 11] and, therefore , ineffective in preventing kernicterus.[12]

Guidelines in several high-income countries advise universal screening for jaundice in newborns using TSB or transcutaneous bilirubin (TcB) measurement.[10 13] Several observational studies have shown a good correlation between TSB and TcB measurements in the neonatal period.[14 15] A programmatic evaluation of a TcB-based screening programme in Canada demonstrated a 55% reduction in severe hyperbilirubinaemia and related healthcare utilisation as compared with visual inspection plus selective TSB quantification.[16] Similar findings were obtained in a retrospective multicentre study evaluating a screening programme using standard versus selective TSB or TcB measurement in the USA.[17] Universal screening for neonatal jaundice in the primary care setting has the potential to reduce the incidence of severe hyperbilirubinaemia.[2 18] This would also bring down the number of patients who require exchange transfusion which is a highly invasive treatment that carries significant health risks and substantial costs.

Hyperbilirubinaemia is usually treated with phototherapy in a hospital setting which may be applied via various modalities.[2] Fibreoptic phototherapy using an underneath device (ie, mattress) is effective in reducing TSB levels, is safe,[19] and meets existing guidelines.[9] There is some evidence to suggest that fibreoptic phototherapy may safely and effectively be used in the primary care setting as well.[14 20 21] The institution of phototherapy in the primary care setting may reduce the number of neonates admitted to hospital for hyperbilirubinaemia treatment and reduce associated costs.[22]

Whereas universal TcB screening for neonatal hyperbilirubinaemia and phototherapy provided in primary care have the potential to be effective and cost-effective via reducing the incidence of severe hyperbilirubinaemia and the need for hospitalisation.[14 23] Evidence to support these assertions is currently lacking, while in today's healthcare arena cost-effectiveness considerations are becoming increasingly important.

We hypothesise that non-invasive screening for neonatal hyperbilirubinaemia using TcB and application of phototherapy in primary care will (cost) effectively reduce the incidence of severe hyperbilirubinaemia and the need for hospital admission for hyperbilirubinaemia treatment. Here, we present our protocol for a factorial stepped-wedge cluster randomised controlled trial (RCT) among newborns cared for in primary care to test these hypotheses.

## METHODS AND ANALYSIS
### Study design
We will conduct a factorial stepped-wedge cluster RCT in seven primary care birth centres (PCBCs) in the Netherlands to evaluate among newborns the effectiveness of (1) screening for hyperbilirubinaemia using daily TcB quantification to reduce the incidence of severe hyperbilirubinaemia and (2) treatment of hyperbilirubinaemia (if indicated) using phototherapy instituted within the PCBC to reduce the need for hospital admission. Accordingly, each PCBC will start with (1) a control phase in which usual care is evaluated, followed by (2) a second phase in which one of both interventions is evaluated, followed by (3) a final phase in which both interventions are evaluated in parallel. More detail is provided below.

### Dutch perinatal care system
In the Netherlands, a significant proportion of women (30% in 2016) give birth in a primary care setting (ie, at home or in a PCBC) under the supervision of a community midwife.[24] In addition, many who deliver in a hospital setting are discharged home or to a PCBC within 24 hours after delivery. Maternity care assistants provide maternity care when the neonate and the mother are at home or in a PCBC, under the supervision of the community midwife (who will visit at least three times in the first week).[25] Only healthy neonates will be cared for at home or in a PCBC (ie, neonates with congenital infection or other diseases will receive initial treatment in the hospital). The community midwife will consult the paediatrician of an affiliated nearby hospital if a potential clinical problem such as hyperbilirubinaemia presents in the neonate. Neonates with a gestational age (GA) above 35 weeks who have been discharged from the hospital are especially at risk of developing severe unconjugated hyperbilirubinaemia.[26]

### Participants
#### Study sites
This study will be performed in seven PCBCs in the Netherlands. In these PCBCs altogether approximately 5000 neonates and their mothers are admitted for at least 48 hours each year. Each of the five largest PCBCs will be considered a cluster in the context of our cluster RCT. The two smallest PCBCs will be paired and will form a cluster together. A PCBC was selected for the study if it facilitated provision of maternity care during the first days post partum (that is, some PCBCs only facilitate deliveries and do not provide care beyond the first 24 hours after delivery).

#### Patient eligibility
Neonates are eligible for inclusion in the study if:
► The neonate was born after 35 completed weeks of gestation. Note: only neonates with a GA of 37 weeks or higher are born in primary care; nevertheless, a neonate may be eligible when born in the hospital between 35 and 37 weeks of GA and then discharged to a PCBC within 7 days after birth.

- ► The neonate is admitted to a participating PCBC within the first week of life.
- ► The neonate is expected to remain admitted for at least 2 days (to allow for serial TcB measurements to take place).
- ► Signed informed consent (IC) is provided by parent(s) or primary caregiver(s).

Neonates are not eligible if:

- ► The neonate received phototherapy previously or is currently receiving phototherapy (reliability of TcB measurement is reduced in neonates who are receiving or have received phototherapy).[14]
- ► Parents do not have sufficient understanding of the Dutch language to be able to comprehend the patient information sheet and questionnaire.

### Interventions and control

In this study, two interventions will be assessed in a 2×2 factorial design to determine the effectiveness of each intervention separately. The timing of the implementation of each intervention will be allocated at the PCBC level.

### Control group: usual care

During this initial study phase, usual care will be evaluated. Accordingly, visual inspection is used to identify jaundice, as advised in the Dutch national guideline for neonatal hyperbilirubinaemia.[9] When potentially severe jaundice is suspected, a skin prick will be performed to determine TSB. The TSB level will be plotted on the Dutch nomogram[27] (which is similar to the nomogram of the American Academy of Pediatrics and explained in box 1) and be discussed with the affiliated paediatrician, according to local procedures. If treatment for hyperbilirubinaemia is indicated, as judged by the affiliated paediatrician, the neonate will be admitted to the hospital for this purpose.

### Intervention 1: TcB measurement as a non-invasive screening tool

During the study phase in which this intervention is tested, TcB will be measured daily starting from 24 hours of age using the Dräger JM-105, a CE-marked and validated hand-held bilirubinometer, in all included neonates admitted to a PCBC allocated to this intervention.[28–31] If TcB exceeds the phototherapy threshold, as determined by the Dutch TSB nomogram for neonates born after 35 weeks of gestation,[9] TSB will be quantified. Because the nomograms were originally designed to be applied to the TSB measurements and because the TcB measurements have an inaccuracy of maximally 50 µmol/L (ie, 2.9 mg/dL),[14 32] TSB will also be quantified if TcB is less than 50 µmol/L below the phototherapy treatment threshold. The TSB level obtained will be discussed with an affiliated paediatrician (of the nearby hospital) in order to determine the need for phototherapy, again based on the Dutch nomogram.[9] A second TcB measurement on the same day may be performed at the discretion

---

**Box 1  The Dutch TSB nomogram[27]**

The Dutch total serum bilirubin (TSB) nomogram has the same values as the nomogram of the American Academy of Pediatrics. The nomogram consists of three risk categories: lower risk, medium risk and higher risk. The risk assessment is based on the gestational age and several risk factors (blood group antagonism, other haemolytic disease, birth asphyxia, suspicion of infection/sepsis, the neonate being ill or drowsy and serum albumin level below 30 g/L). The need for phototherapy and exchange transfusion depend on the risk assessment and the postnatal age of the neonate. A translated version of the Dutch TSB nomogram is depicted in figure 1.

of the maternity care professional (eg, if the TcB level is close to the threshold for measuring TSB).

Following institution of phototherapy, bilirubin levels will be monitored using TSB rather than TcB given the reduced reliability of TcB in neonates receiving or having received phototherapy.[14]

### Intervention 2: phototherapy in the PCBC

During this study phase, the default location for application of phototherapy, if clinically indicated, will be the PCBC, rather than the hospital. The indication for phototherapy is made by a paediatrician in the affiliated hospital based on the TSB level, according to the Dutch nomogram.[9] Phototherapy in the PCBC will be provided using GE Healthcare BiliSoft Large, a commercially available, CE-marked and validated phototherapy mattress using blue LED-light.[33] This phototherapy mattress is suitable for application in primary care as well as in the hospital setting.

The affiliated paediatrician will decide whether other blood tests have to be performed to determine the cause of hyperbilirubinaemia and at what time points follow-up TSB measurements need to be taken. Based on the results of these TSB measurements (dis)continuation of phototherapy and further follow-up is determined by the affiliated paediatrician. The affiliated paediatrician may decide to admit a neonate to the hospital for further hyperbilirubinaemia treatment at any time if there are strong reasons to do so; these reasons will be recorded. Phototherapy will not be applied in the PCBC in neonates who have hyperbilirubinaemia above the phototherapy threshold within 24 hours after birth, since this is considered to represent a likely pathological cause of hyperbilirubinaemia and bilirubin levels may rise swiftly in these neonates.[9]

Each PCBC will start with a time period without any intervention (control period), followed by a time period with only one of the interventions, and then a time period with both interventions. The inclusion of neonates will take 2.5 years per PCBC. The sequence and the timing at which the PCBC will switch over to the next intervention period will be determined by random allocation. The allocation scheme with the possible sequences of the interventions is depicted in figure 2. The time blocks will take 4 months, except for the first and the last block.

---

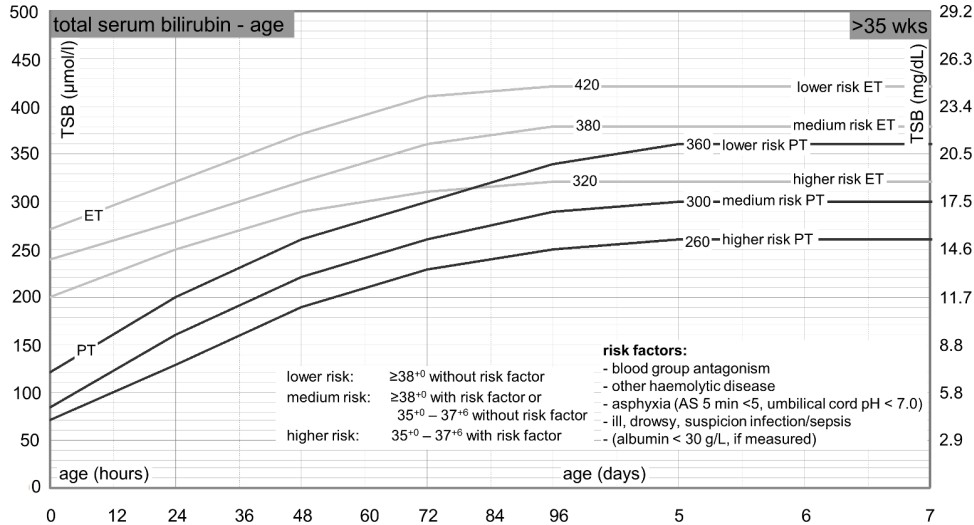

**Figure 1** Phototherapy and exchange transfusion thresholds for neonates with a gestational age of 35 weeks or more. Translated from Dutch. The Dutch nomogram can be downloaded from the website (http://babyzietgeel.nl/kinderarts/hulpmiddelen/diagnostiek/bilicurve35wkn.php). AS, Apgar score; ET, exchange transfusion; PT, phototherapy; TSB, total serum bilirubin.

These two blocks will cover 5.5 months, because, to be able to make a comparison within a cluster, a sufficient number of participants in the control period (C) and in the period with both interventions (I1 + I2) should be included, even when a smaller PCBC is randomised to time line 1 or 6.

A flowchart with an overview of the study procedures with the different intervention phases and possible orders of intervention phases from the perspective of a participant is displayed in figure 3.

### Primary outcomes
The two interventions each have their own primary outcome. The primary outcome for TcB measurement as a screening tool for jaundice will be the proportion of neonates having severe hyperbilirubinaemia at any time point within the first 14 days of life. Severe hyperbilirubinaemia is defined as a TSB level above the mean of the phototherapy and exchange transfusion thresholds, according to the Dutch nomogram[9];that is,

$$TSB > \frac{phototherapy\ threshold + exchange\ transfusion\ threshold}{2}$$

For the second intervention, phototherapy in the PCBC, the primary outcome will be the proportion of neonates requiring hospital admission for hyperbilirubinaemia

treatment within the first 14 days of life. Hyperbilirubinaemia treatment will be considered the indication for hospital admission if the neonate received phototherapy or an exchange transfusion for hyperbilirubinaemia within 24 hours after hospital admission.

### Secondary outcomes
The secondary outcomes will also be assessed within the first 14 days of life (except for kernicterus) and are listed in table 1.

### Cost-effectiveness analysis
A cost-effectiveness analysis (CEA) will be performed from the healthcare perspective. In the CEA, the costs and outcomes of the interventions, compared with 'usual care', will be identified, measured and valued.

For all of the following items, healthcare use will be measured: TSB and TcB tests, phototherapy in the hospital, phototherapy in the PCBC, exchange transfusions, PCBC and/or hospital admission days and ambulance transportation to a hospital. These data will be recorded during the study and/or retrieved from the electronic information systems of the participating PCBCs and affiliated hospitals.

Then, integral unit prices will be calculated using real economic cost prices or using standard cost-prices for health economic evaluations.[34] Unit prices will be multiplied by the quantities for each resource used and summed over the separate types of resources to give the total cost per patient.

Regarding the outcomes, the CEA will look at the number of neonates with severe hyperbilirubinaemia (as defined above) and the number of hospital admissions. Data on costs and outcomes will be measured throughout the 14-day observation period. As the final outcome measure of the CEA, incremental cost effectiveness ratios

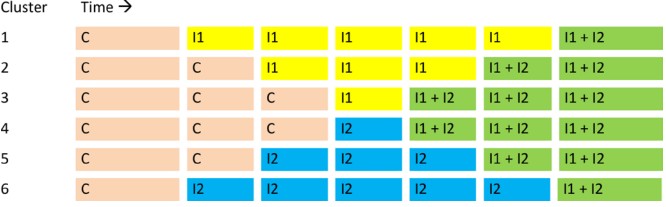

**Figure 2** Allocation scheme. C, control; I1, intervention 1, transcutaneous bilirubinometry measurement; I2, intervention 2, phototherapy in primary care birth centres.

van der Geest BAM, et al. BMJ Open 2019;9:e028270. doi:10.1136/bmjopen-2018-028270

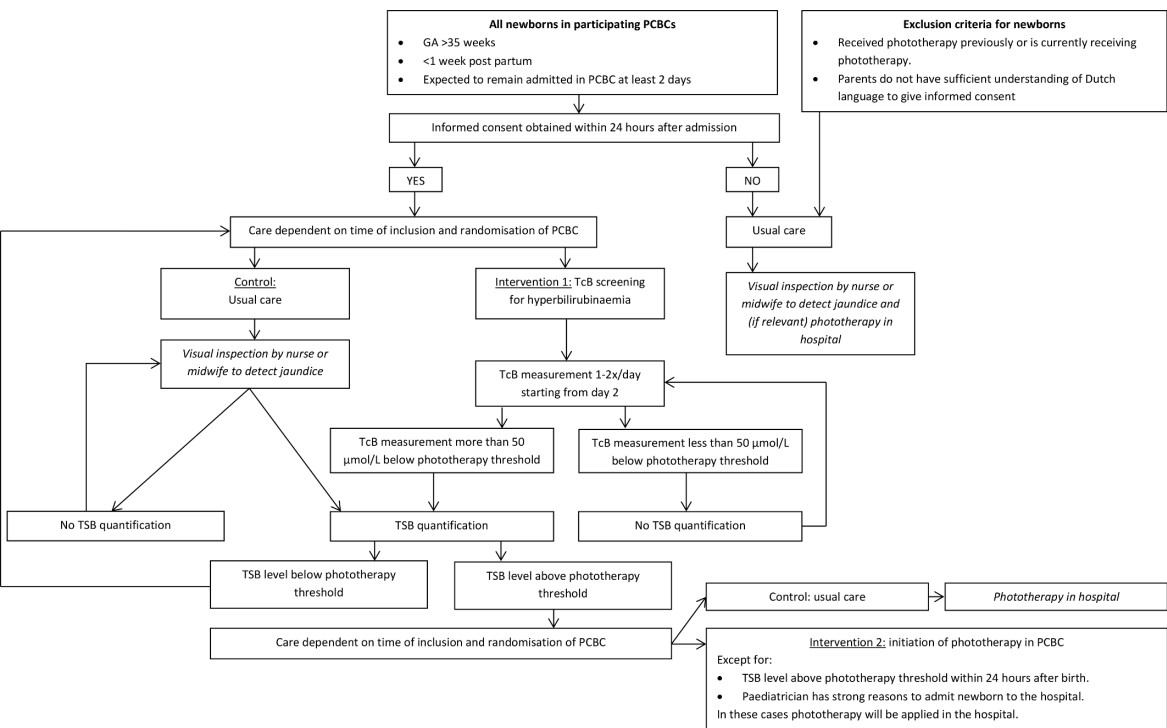

**Figure 3** Flowchart for study inclusion. GA, gestational age; TcB, transcutaneous bilirubinometry; TSB, total serum bilirubin; PCBC, primary care birth centres.

(ICERs) will calculated, expressed as incremental costs per case of severe hyperbilirubinaemia or hospitalisation avoided. Uncertainty in the estimation of the ICERs will be illustrated through cost-effectiveness planes. Where relevant, sensitivity analyses will be performed to assess the robustness of the analyses to certain assumptions.

## Sample size

Two sample size calculations are required since two interventions, each with their own outcomes, are evaluated. The interventions are considered to be independent of each other and therefore the larger sample size among the two sample size calculations is considered appropriate for the entire study. Sample size calculations are based on detecting superiority rather than non-inferiority.

The estimated incidence of severe hyperbilirubinaemia in term neonates in the Netherlands is 1.2% per year.[9] A previous study in Canada regarding screening for unconjugated hyperbilirubinaemia using TcB among healthy neonates with ≥35 weeks of gestation showed a relative reduction of 55% in severe hyperbilirubinaemia following implementation.[16] We therefore hypothesise that the TcB screening intervention will decrease the incidence of severe hyperbilirubinaemia from 1.2% to 0.5% (ie, by 0.7% points; a relative reduction of 58%). At a power of 80% and a two-sided alpha of 0.05, 2691 neonates per arm are needed to identify this effect.

The estimated proportion of neonates needing phototherapy is 4%.[9] We anticipate that approximately 50% of the neonates with hyperbilirubinaemia will have an indication to be admitted to the hospital (eg, feeding difficulties in addition to hyperbilirubinaemia or need for

intensive phototherapy). Therefore, we hypothesise that the planned application of phototherapy in the PCBC will decrease the need for hospital admission for hyperbilirubinaemia by 2% points (ie, 50% relative reduction). At a power of 80% and a two-sided alpha of 0.05, 1136 neonates per arm are needed to identify this effect. While our study is a cluster RCT, given the cross-sectional nature of a stepped-wedge design the effect of intracluster correlation on power is minimal, therefore we did not adjust for intracluster correlation.[35]

Based on the larger sample size of 2691 neonates per arm, 5382 neonates are needed in total. We aim to include 5500 neonates to account for a degree of missing data.

## Randomisation and blinding

Each of the participating PCBCs will be randomised to one of the predefined timelines (as depicted in figure 2 above) by an online randomisation module (www.random-ization.com). The two smallest PCBCs will be paired and randomised to the same timeline (see figure 2).

Given the nature of the treatment in the intervention groups and the control group, blinding of participants (and parents) and healthcare personnel is not possible. Analyses will be performed by a researcher who is blinded to the PCBC allocation scheme.

## Study procedures

All PCBCs in the Netherlands that provide maternity care to mothers and neonates in the first days after delivery were invited to participate. Eligibility of neonates admitted to the PCBC will be assessed by the maternity care or study personnel based on the inclusion and

**Table 1** Secondary outcomes

| Secondary outcomes | | |
| --- | --- | --- |
| **Assessed in** | **Outcome** | **Recorded by** |
| All neonates | Number of times TcB quantified | Maternity care professional |
| | Individual TcB readings | Maternity care professional |
| | Highest TSB level | Maternity care professional and/or EMR in hospital |
| | Number of neonates having TSB quantified | Maternity care professional |
| | Number of times blood taken for TSB quantification | Maternity care professional and/or EMR in hospital |
| | Individual TSB levels | Maternity care professional and/or EMR in hospital |
| | Number of times blood taken for TSB quantification before start of phototherapy | Maternity care professional and/or EMR in hospital |
| | Number of neonates receiving phototherapy | Maternity care professional and/or EMR in hospital |
| | Duration (hours) of phototherapy | Maternity care professional and/or EMR in hospital |
| | Number of neonates having TSB level above exchange transfusion threshold[9] | Maternity care professional and/or EMR in hospital |
| | Number of neonates who actually received an exchange transfusion | EMR in hospital |
| | Number of neonates having kernicterus* | EMR in hospital and/or EMR at general practitioner |
| | Duration of stay in the PCBC | Maternity care professional |
| | Duration of hospital stay following initial PCBC admission (if relevant) | EMR in hospital |
| | Number of transfers between PCBCs/hospitals | Maternity care professional and parental questionnaire |
| | Number of times affiliated paediatrician consulted | Maternity care professional |
| | Experience of parents regarding hyperbilirubinaemia assessment and treatment | Parental questionnaire |
| | Experience of attending maternity care personnel regarding the daily practice of hyperbilirubinaemia assessment and treatment during control and intervention periods, including facilitators and barriers for implementation | Maternity care assistant questionnaire |
| Within the group included in the intervention period with phototherapy in the PCBC | The number of neonates in whom phototherapy was initiated in the hospital, including reasons for this hospital admission | EMR in hospital |
| | The number of neonates requiring subsequent hospital admission for hyperbilirubinaemia treatment (ie, after initiation of phototherapy in the PCBC), including reasons for this 'treatment failure' | EMR in hospital |

*The diagnosis of kernicterus will be made by combining clinical signs and symptoms and additional investigations (eg, cerebral ultrasound, MRI) up to 1 year after birth in every neonate who had a TSB level above the exchange transfusion threshold level.
EMR, electronic medical record; PCBC, primary care birth centres; TcB, transcutaneous bilirubinometry; TSB, total serum bilirubin.

exclusion criteria. IC of parents will be obtained by the maternity care professional. Research nurses and study personnel will support the maternity care professionals in obtaining IC and collecting data.

Depending on the allocation and the moment of admission in the PCBC during the study period, the neonate is included in the control period, in a time period where one of the interventions is implemented, or in a time period where both interventions are implemented. Standard operating procedures (SOPs) for each study phase have been developed.

Maternity care personnel of all PCBCs will be trained in the SOPs and in using the transcutaneous bilirubinometer and the phototherapy mattress to ensure the interventions are applied in a professional standardised manner. For this purpose, an e-learning

| Table 2 Baseline characteristics | |
| --- | --- |
| **Baseline characteristics** | |
| Parental characteristics | Maternal age |
| | Maternal ethnicity |
| | Maternal blood group |
| | Paternal ethnicity |
| | Presence of haemolytic disease (other than blood group antagonism) in mother or father |
| Characteristics of pregnancy and birth | Parity |
| | Maternal atypical red-cell alloantibodies during pregnancy |
| | Gestational age at birth |
| | Type of delivery: vaginal delivery, vacuum-assisted vaginal delivery, forceps-assisted vaginal delivery, caesarean delivery without vacuum or forceps extraction (before or during delivery), caesarean delivery with vacuum or forceps extraction (before or during delivery). |
| Neonatal characteristics | Date of birth |
| | Time of birth |
| | Sex |
| | Birth weight (in grams) |
| | Presence of birth trauma |
| | Type of feeding |
| | Neonatal blood group (if known) |
| | Foetal or neonatal Rhesus factor (if known) |
| | Direct antiglobulin (Coombs) test (if known) |
| | Presence of haemolytic disease (other than blood group antagonism) |
| | Presence of birth asphyxia: Apgar score <5 at 5 min or pH <7.0 in umbilical cord blood |
| | Suspicion of sepsis |
| | Neonate being ill or drowsy |
| Other characteristics | Siblings who experienced neonatal hyperbilirubinaemia (and cause of hyperbilirubinaemia, if known) |

module has been developed (available from: https://xoteur.12change.eu/play.php?template_id=798); in addition, during the study period three training sessions for maternity care professionals are held at each PCBC prior to the start of each subsequent study phase (ie, intervention implementation).

## Data collection

Data collection at the PCBC will be performed by maternity care personnel. Together with the training for using the transcutaneous bilirubinometer and the phototherapy mattress, maternity care personnel will be trained in adequate data collection. Data collection is described in the SOPs as well.

After inclusion, a maternity care professional of the PCBC will record baseline characteristics of the mother and the neonate (see table 2) using a standardised registration form at a secured internet page. All participants will receive an anonymised study number.

The maternity care professionals will record several measurements during PCBC stay. These measurements are noted in table 3. All data will be recorded in standardised case report forms. The study team, the study monitor, and (inter)national supervisory authorities will have access to the data and the final dataset.

A questionnaire for the parents/caregivers to evaluate their experience will be sent by email 14 days after the birth of the neonate. This questionnaire will assess parental satisfaction, burdensomeness of interventions for the neonate and experienced competence of maternity care professionals. If parents do not have access to the internet or do not respond to the questionnaire by email, they will be approached by regular mail or by phone. The community midwife and/or the general practitioner will be approached if parents do not respond at all. Both after the control and after each intervention period, maternity care personnel will be invited to complete a questionnaire about their experiences regarding the daily practice of hyperbilirubinaemia assessment and treatment during these periods (eg, facilitators and barriers for implementation, sense of competence regarding applying the new interventions, integrating the new interventions in routine care, and the content of the training regarding the new interventions).

## Statistical analysis

We will use cluster-specific methods because randomisation will be performed at the PCBC level rather than at

**Table 3** Daily measurements in PCBC

| Daily measurements | |
|---|---|
| Control period | Skin colour: pink, slightly yellow, moderately yellow, quite yellow, very yellow |
| | Weight (in grams) |
| | Risk factors for hyperbilirubinaemia (if present) |
| | TSB values with date and time of measurement (if relevant) |
| | Decisions made based on TSB (if relevant) |
| Period with TcB screening (extra measurements in addition to control period) | TcB values measured at forehead and sternum together with date and time of measurement (once a day or two times if indicated) |
| Phototherapy in PCBC (extra measurements in addition to control period) | Start and end date and time of phototherapy |
| | Decisions made by the affiliated paediatrician regarding phototherapy |

PCBC, primary care birth centres; TcB, transcutaneous bilirubinometry ; TSB, total serum bilirubin.

the individual level. For both primary objectives, an intention-to-treat analysis will be performed.

To evaluate whether the occurrence of the primary outcomes of both interventions are different between the study periods with and without the intervention(s), generalised linear mixed models (GLMMs) will be used. GLMMs account for the systematically different observation periods and for clustering in the data.[36] In case of missing data, multiple imputation using chained equations will be used. Relative risks, risk differences, and relative risk reduction with corresponding 95% CIs will be calculated from the GLMMs.

Subgroup analyses to evaluate the effect of the interventions in neonates of different ethnicities (Caucasian and non-Caucasian) and different GA groups (<38 weeks of gestation and ≥38 weeks of gestation) will be performed.

We will perform a per-protocol analysis to assess the impact of potential contamination during the first days of a new intervention period as a sensitivity analysis in addition to the intention-to-treat analysis.

### Safety
Adverse events noticed by parents or maternity care professionals will be recorded in the case report forms by the maternity care assistant or study personnel. Serious adverse events will be communicated to the study team within 4 days. The study team will investigate this event and report it to the sponsor and the Medical Research Ethics Committee of the Erasmus MC Rotterdam. Insurance has been taken out for every participant in this study. The insurance covers any damage caused by the study.

The medical technical service department of the Erasmus MC-Sophia will provide technical maintenance of the phototherapy mattresses (at least two times per year). Once a year, calibration of the TcB devices will be carried out by the manufacturer.

A Data Safety Monitoring Board will not be established, given that the risks of this study are considered negligible. Monitoring of the study will be carried out according to the Guideline for Good Clinical Practice by an independent, qualified monitor at least once a year per PCBC.[37]

### Patient and public involvement
The study protocol, study procedures and patient information form were discussed with the regional patient involvement board of the Rijnmond Obstetric District Platform ('District Verloskundig Platform'; DVP) and patient representatives of the Child & Hospital foundation ('Stichting Kind & Ziekenhuis': https://www.kindenziekenhuis.nl/). They consider the project highly relevant from a patient perspective and have agreed to be involved throughout execution, analysis and interpretation, dissemination to patients and implementation. The involved patient panels have considered the burden of the interventions negligible and underline the potential benefits of the study.

### Ethics and dissemination
Collected data will be coded. Only investigators will have access to the identification list. Study documents and data will be stored for 15 years after completion of the study.

The full protocol will be shared at request. Results of this study including the statistical code will be published in peer reviewed scientific journals and be presented at (inter)national meetings. The results of the study will be reported according to the Consolidated Standards of Reporting Trials Statement guidelines.[38] We intend to provide an anonymised version of the dataset at request. Authorship will be determined according to the guidelines of the International Committee of Medical Journal Editors.[39] No professional writers will be used.

### DISCUSSION
Identification and (referral for) treatment of neonatal jaundice is daily practice for maternity care professionals. The continuing occurrence and devastating consequences of KSD underline the urgent need for more effective identification and treatment of at-risk neonates.[1 7] This pragmatic study aims to evaluate the (cost) effectiveness

of (1) TcB measurement as a screening tool for jaundice in a primary care setting and (2) application of phototherapy in the PCBCs in the Netherlands. To the best of our knowledge, this RCT will be the first to assess the (cost) effectiveness of these diagnostic and treatment modalities in a primary care setting.

The study design allows all participating PCBCs to implement both interventions during the study and as such all PCBCs will have implemented the interventions as standard care toward the end of the study. This process will provide valuable information about the facilitators and barriers for implementation that will help upscaling of the interventions if (cost)effectiveness is confirmed.

A limitation of this study is that parents and care providers cannot practically be blinded to allocation of the interventions. As a result, maternity care personnel or paediatricians may act differently in control and intervention periods. We will try to identify this via the questionnaires addressed at the maternity care assistants and the recording of the underlying reasons for paediatricians to divert from the new practice of applying phototherapy in the PCBC, if relevant. We will, furthermore, record the number of times TSB is quantified to evaluate whether this may change following implementation of the screening programme.

For this study we take advantage of the fact that neonates receive primary care within PCBCs rather than at home. This centralised primary care provision is efficient in terms of feasibility of training the maternity care professionals and of minimising costs of transcutaneous bilirubinometers and phototherapy mattresses (ie, groups of neonates can be covered with individual devices). Moreover, although the Dutch perinatal healthcare system is quite unique, care provision in PCBCs in the Netherlands is comparable to regular maternity care in many other high-income countries, which promotes generalisability.

We expect the interventions to be similarly effective in the home setting as in the PCBC setting, since the population in the PCBCs in the Netherlands does not differ from that of the home setting. The costs in the PCBC setting may however differ from the home setting. Generalisability of the results to other countries and the home settings depends on several aspects, such as characteristics of the population, organisation of the maternity care system, presence of an effective screening programme for neonatal hyperbilirubinaemia and proximity and availability of sufficient healthcare. A future study to investigate the feasibility and effectiveness of the interventions in the home setting is needed if the current study is successful.

In conclusion, our study will generate useful data about the (cost)effectiveness of TcB measurement as a universal screening tool for jaundice, potentially reducing the incidence of severe neonatal hyperbilirubinaemia. Several guidelines have highlighted the lack of research into the prevention of severe hyperbilirubinaemia.[9 10 13 40] Although most of these guidelines do not specify where phototherapy should be instituted, it is standard practice in most high-income countries, including the Netherlands, to do so in the hospital setting.[9 10 40] As such, institution of phototherapy in the primary care setting has significant potential to reduce healthcare costs associated with hospital admission. Furthermore, it promotes mother-child bonding by allowing mother and child to remain together in the same location.

**Author affiliations**
[1]Division of Neonatology, Department of Paediatrics, Erasmus MC - Sophia Children's Hospital, University Medical Centre Rotterdam, Rotterdam, Netherlands
[2]Department of Obstetrics and Gynaecology, Erasmus MC - Sophia Children's Hospital, University Medical Centre Rotterdam, Rotterdam, Netherlands
[3]Medical Technology Assessment (iMTA), Erasmus University Rotterdam, Rotterdam, Netherlands
[4]Paediatric Intensive Care Unit, Department of Paediatrics, Erasmus MC - Sophia Childen's Hospital, University Medical Centre Rotterdam, Rotterdam, Netherlands
[5]Paediatric Surgery, Erasmus MC - Sophia Children's Hospital, University Medical Centre Rotterdam, Rotterdam, Netherlands
[6]Nursing Science, Department of Internal Medicine, Erasmus MC, University Medical Centre Rotterdam, Rotterdam, Netherlands
[7]Department of Public Health, Erasmus MC, University Medical Centre Rotterdam, Rotterdam, Netherlands

**Correction notice** This article has been corrected since it first published online. The open access licence type has been amended.

**Acknowledgements** We acknowledge The Netherlands Organisation for Health Research and Development and Erasmus MC for their financial support to this research. We thank Rogier de Jonge for his support concerning the design and funding of this study, Verena Werdmüller von Elgg for her support regarding Medical Research Ethics Committee approval, Liesbeth Dusink and Roel Faber for their support regarding preparation for data collection and composing case report forms, Marc Sylva for help with literature review, Nina Witt, Rik Bouman and Jetty van Ginkel for their support regarding legal issues and contact with manufacturers, Imke Theeuwen for support during the training sessions, and other practical issues in the phase before the start of the study. We thank the patient involvement board of the Rijnmond Obstetric District Platform and the patient representatives of the Child & Hospital Foundation for their feedback regarding the patient perspective. We are grateful to the participating PCBCs, maternity care professionals, midwives and affiliated paediatricians for taking part in this study.

**Collaborators** On behalf of the STARSHIP study group: Martin GA Baartmans (Department of Paediatrics, Maasstad Hospital, Rotterdam, The Netherlands), Jolita Bekhof (Department of Paediatrics, Isala – Amalia Children's Clinic, Zwolle, The Netherlands), Harry Buijs (Primary care birth centre Haga, The Hague, The Netherlands; Primary care birth centre Maasstad, Rotterdam, The Netherlands), Jan Erik Bunt (Department of Paediatrics, Elisabeth-TweeSteden Hospital, Tilburg, The Netherlands), Peter H Dijk (Department of Neonatology, University Medical Centre Groningen – Beatrix Children's Hospital, University of Groningen, Groningen, The Netherlands), Marja C Huizer (Primary care birth centre Haga, The Hague, The Netherlands; Primary care birth centre Maasstad, Rotterdam, The Netherlands; Primary care birth centre Rotterdam Noord, Rotterdam, The Netherlands), Christian V Hulzebos (Department of Neonatology, University Medical Centre Groningen – Beatrix Children's Hospital, University of Groningen, Groningen, The Netherlands), Ralph WJ Leunissen (Department of Paediatrics, Haaglanden Medical Centre Westeinde, The Hague, The Netherlands), Beata Pazur (Primary care birth centre Rotterdam Noord, Rotterdam, The Netherlands), Ben JPW Snoeren (Primary care birth centre Fam, Tilburg, The Netherlands), Bente de Vries (Primary care birth centre Westeinde, The Hague, The Netherlands), Leo Wewerinke (Department of Paediatrics, Haga Hospital – Juliana Children's Hospital, The Hague, The Netherlands).

**Contributors** BAMvdG and JVB wrote the first version of the manuscript. JVB, JPdG, RFK, IKMR and EAPS designed the study and secured funding. JVB and LCMB developed the methodological part of this study. MJP designed the cost-effectiveness analysis. EI designed evaluation of implementation of both interventions. All authors critically revised the manuscript and gave final approval of the version to be published.

**Funding** This work is supported by a Health Care Efficiency Research grant of The Netherlands Organisation for Health Research and Development (ZonMw), grant

number 843002805, and an Erasmus MC Efficiency Research grant, grant number 2016-16107. The study funders were not involved in the collection, management, analysis and interpretation of data; writing of the report; or the decision to submit the report for publication.

**Competing interests** None declared.

**Patient consent for publication** Written IC is provided by parents of neonates included in the study.

**Ethics approval** This study has been approved by the Medical Research Ethics Committee of the Erasmus MC Rotterdam (MEC- 2017 – 473) .

**Provenance and peer review** Not commissioned; externally peer reviewed.

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
