## [Reviewer comments · BMJ Open]

ARTICLE DETAILS

TITLE (PROVISIONAL)	Screening and TreAtment to Reduce Severe Hyperbilirubinaemia in Infants in Primary care (STARSHIP): a factorial stepped wedge cluster randomised controlled trial protocol
AUTHORS	van der Geest, Berthe; de Graaf, J; Bertens, Loes; Poley, Marten; Ista, Erwin; Kornelisse, René; Reiss, Irwin; Steegers, Eric; Been, JV; STARSHIP study group

VERSION 1 - REVIEW

REVIEWER	Ichiro Morioka Department of Pediatrics and Child Health, Nihon University School of Medicine, Tokyo, Japan
REVIEW RETURNED	19-Jan-2019

GENERAL COMMENTS	Dr. van der Geest et al. planned a randomized controlled trial whether, among babies cared for in primary care: 1. transcutaneous bilirubinometry (TcB) screening can help reduce severe hyperbilirubinaemia, and 2. primary care-based versus hospital-based phototherapy can help reduce hospital admissions. In addition, a cost-effectiveness of the interventions will be investigated. I can understand the contents of this study. The study design and protocol will be agreed. I have one comment for the modification. Please describe more precisely regarding the exclusion patients. Although authors have described that patients with pathological jaundice are excluded, can congenital anomalies, chromosomal abnormality, or congenital infection and so on be included? Perhaps, as most of these patients will be admitted in a hospital, I expect most of these will be the exclusion. However, if these patients will be in primary care facility, how about it?
--

REVIEWER	Bolajoko Olusanya FRCPCH, PhD Centre for Healthy Start Initiative Lagos, Nigeria
REVIEW RETURNED	28-Jan-2019

GENERAL COMMENTS	The authors plan to test the composite hypothesis that that non-invasive screening for neonatal hyperbilirubinaemia using TcB and phototherapy in primary care will be cost-effective in reducing the incidence of severe hyperbilirubinaemia and the need for hospital admission for hyperbilirubinaemia treatment using a factorial stepped-wedge cluster randomised controlled trial (RCT). The
--

	protocol is well laid out and the findings should be of interest to clinicians in neonatology practice. Minor issues to address:  1. The planned enrolment of participants as reported the manuscript is between July 2018 and July 2021. Has enrolment already started? 2. In the light of the authors' prior study in 2017, what plans (if any) will be made to ensure that the phototherapy devices are adequately effective and maintained throughout the duration of the trial? 3. Page 22, Line 22: add "in a primary-care setting" after ".....treatment modalities." 4. I will caution against generalising the findings in the primary care centres to homes, regardless of the similarity in the population. A future study may explore the feasibility of home phototherapy compared to the primary care centres. Even then, facility-based treatment should be recommended because of the potential risk of kernicterus. 5. An express statement in the manuscript that the study will comply with CONSORT should be considered.
--	---

VERSION 1 – AUTHOR RESPONSE

REVIEWER 1

Please describe more precisely regarding the exclusion patients. Although authors have described that patients with pathological jaundice are excluded, can congenital anomalies, chromosomal abnormality, or congenital infection and so on be included? Perhaps, as most of these patients will be admitted in a hospital, I expect most of these will be the exclusion. However, if these patients will be in primary care facility, how about it?

Response: In principle, only healthy neonates will be cared for in a primary care setting. Neonates with significant congenital anomalies, chromosomal abnormalities, or congenital infections will initially be admitted to the hospital. If such diseases appear in a participant of the study, the participant will be admitted to the hospital and information on the hospital admission will be retrieved from the general practitioner and/or the hospital. These diseases are not an exclusion criterion as such, since this is according to daily practice in the primary care birth centres and these neonates may develop neonatal hyperbilirubinaemia as well.

REVIEWER 2

1. The planned enrolment of participants as reported the manuscript is between July 2018 and July 2021. Has enrolment already started?

Response: The enrolment has already been started. We have changed this (Patient eligibility, page 9).

2. In the light of the authors' prior study in 2017, what plans (if any) will be made to ensure that the phototherapy devices are adequately effective and maintained throughout the duration of the trial?

Response: The medical technical department of the Erasmus MC-Sophia will perform check-ups every six months on the phototherapy mattresses. We have provided information on this topic (Safety, page 21).

3. Page 22, Line 22: add “in a primary-care setting” after “.....treatment modalities.”

Response: We have added this accordingly.

4. I will caution against generalising the findings in the primary care centres to homes, regardless of the similarity in the population. A future study may explore the feasibility of home phototherapy compared to the primary care centres. Even then, facility-based treatment should be recommended because of the potential risk of kernicterus.

Response: We are aware of the differences between a primary care birth centre and the home setting and agree that one should be cautious in extrapolating findings from this study to the home situation. We added the suggestion of a feasibility study (Discussion, page 23).

5. An express statement in the manuscript that the study will comply with CONSORT should be considered.

Response: This has been added to the manuscript (Ethics and dissemination, page 22).

VERSION 2 – REVIEW

REVIEWER	Bolajoko Olusanya Executive Director Centre for Healthy Start Initiative Ikoyi, Lagos Nigeria
REVIEW RETURNED	15-Feb-2019

GENERAL COMMENTS	No additional comments.
-------------------------